# Impact of Mogamulizumab in Real-Life Advanced Cutaneous T-Cell Lymphomas: A Multicentric Retrospective Cohort Study

**DOI:** 10.3390/cancers14071659

**Published:** 2022-03-25

**Authors:** Marie Jouandet, Inès Nakouri, Lawrence Nadin, Alice Kieny, Mahtab Samimi, Henri Adamski, Gaëlle Quéreux, Guillaume Chaby, Anne Dompmartin, Jean-Matthieu L’Orphelin

**Affiliations:** 1Department of Dermatology, Caen-Normandie University Hospital, 14000 Caen, France; jouandet-m@chu-caen.fr (M.J.); nakouri-i@chu-caen.fr (I.N.); dompmartin-a@chu-caen.fr (A.D.); 2Biostatistics and Clinical Research Unit, Caen-Normandy University Hospital, 14000 Caen, France; nadin-l@chu-caen.fr; 3Department of Dermatology, Strasbourg Civil Hospital, 67000 Strasbourg, France; alice.kieny@chru-strasbourg.fr; 4Department of Dermatology, Centre Hospitalier Régional et Universitaire de Tours, 37000 Tours, France; mahtab.samimi@univ-tours.fr; 5Department of Dermatology, Centre Hospitalier Universitaire Ponchaillou, 35000 Rennes, France; henri.adamski@chu-rennes.fr; 6Department of Dermatology, Nantes University Hospital, 44000 Nantes, France; gaelle.quereux@chu-nantes.fr; 7Department of Dermatology, Amiens-Picardie University Hospital, 80000 Amiens, France; chaby.guillaume@chu-amiens.fr

**Keywords:** cutaneous T-cell lymphoma, adverse events, mogamulizumab, Sézary, mycosis fungoides

## Abstract

**Simple Summary:**

Mogamulizumab is a recent monoclonal antibody prescribed in the second line to treat advanced mycosis fungoides and Sézary syndromes. We collected data from all patients who used mogamulizumab in six French university hospitals until 1 September 2021. Our primary objective was to determine the median progression free survival (PFS). Secondary objectives were to consider tolerance regarding side effect occurrence and severity. Twenty-one patients were included, with a median time of follow-up of 11.6 months, and progression-free survival was estimated at 22 months. Twenty patients presented adverse events, of which 10 were severe. The median time between the introduction of mogamulizumab and the first adverse event was 21 days. Our study suggests that mogamulizumab is a significant treatment option to extend PFS in patients with advanced refractory cutaneous T-cell lymphomas (CTCL). The long-term safety of mogamulizumab was determined to be acceptable since we reported few grade III–IV adverse events (AEs) compared to other systemic treatments.

**Abstract:**

Background: Advanced mycosis fungoides (MF) and Sézary syndrome (SS) are rare, aggressive cutaneous T-cell lymphomas that may be difficult to treat. Mogamulizumab is a recent monoclonal antibody targeting the CCR4 receptor expressed on the surface of Sézary cells. It can be prescribed in MF/SS stages III to IV in the second line after systemic therapy or in stages IB-II after two unsuccessful systemic therapies. We lack data on long-term efficiency and potential side effects in real-life conditions. Our study aims to determine efficacy considering the median PFS of advanced CTCL with mogamulizumab. Secondary objectives were to consider tolerance and estimate delay until side effects appeared. Methods: Data on patients with advanced cutaneous T-cell lymphomas were collected since French Authorization, in six French university hospitals. Patients were followed until they stopped mogamulizumab because of relapse or toxicity. For those still treated by mogamulizumab, the end point was 1 September 2021. We excluded 3 patients as they had already been included in the MAVORIC study and data was not available. Results: The median time of follow-up was 11.6 months. Of the 21 patients included, we reported four full-response patients, eight in partial response, one in stability, three in progression, and five were deceased. One patient had visceral progression, and seven had new lymphadenopathy. Progression-free survival was estimated at 22 months. Twenty patients presented adverse events, of which 10 were severe, i.e., grade III-IV. The median time between the introduction of mogamulizumab and the first adverse event was 21 days. Conclusions: Our study suggests that mogamulizumab can give patients with advanced refractory CTCL a consequent PFS, estimated at 22 months. The long-term safety of mogamulizumab was determined to be acceptable since we reported few grade III–IV AEs, comparable with other studies. No other study using real-life data has been performed to investigate the AEs of mogamulizumab.

## 1. Introduction

Lymphomas are mostly malignant tumors that develop within lymphoid tissues [1]. Primary cutaneous lymphomas are defined by lymphomas affecting the skin at diagnosis without extra-cutaneous involvement. Among primary cutaneous lymphomas, we can separate cutaneous T-cell lymphomas (CTCL) and cutaneous B-cell lymphomas, depending on whether they affect T-lymphocytes or B-lymphocytes.

CTCL are rare diseases. They represent 2% of non-Hodgkin lymphomas but also 83% of cutaneous lymphomas [2]. The incidence rate of CTCL has probably increased over the past 30 years to reach 2.9 and 7.7 new cases per year per million inhabitants, respectively, in Europe [3] and the USA [4]. It affects men twice as frequently as women. It occurs at any stage of life, even childhood, but the incidence increases significantly after 40 years old, and most diagnoses are established after 60 years old [5].

CTCL can be distinguished into two main groups with different evolution and prognostic profiles. On the one hand, mycosis fungoides (MF) and the following well-defined (WHO-EORTC) [6] variants: pagetoid reticulosis, granulomatous slack skin, and folliculotropic, excluding erythrodermic MF. It is the most common expression of CTCL, presenting, in most cases, an indolent evolution with a good prognosis. On the other hand, erythrodermic presentations of MF and Sézary syndrome (SS) are less common and have a more pejorative prognosis.

MF is the most common stage of primary cutaneous T-cell lymphomas. Elementary lesions, mostly itchy, are patches (non-infiltrative erythematous lesions, T1), plaques (infiltrative erythematous lesions, T2), and then tumors (T3). In histology, it is defined by an epidermotropic infiltrate of CD4^+^ T-cells (lymphocytes). It is estimated that 20 to 25% of patients will progress to an advanced stage, defined by the presence of tumors, erythroderma, and visceral or lymphatic damage [7].

SS is a rare disease (5% of CTCL). It is an aggressive and leukemic form of CTCL in which there is erythroderma, lymphadenopathy, and a high level of atypic T-cell lymphocytes with cerebriform nuclear contour called Sézary cells.

MF and SS affect quality of life through visible lesions, chronic itching, anxiety, or social issues [8]. Advanced MF and SS are aggressive forms of lymphomas as the median overall survival is around 5 years [9]. On histology, these two types of lymphomas are characterized by a lymphocytic infiltrate with cerebriform nuclei and a haloed appearance that display epidermotropism or populate the dermo-epidermal junction [10].

Flow cytometry is a sophisticated technique for measuring physical characteristics of a cell [11] and allows the recognition of some specific cells, particularly the expression of CD markers by specific monoclonal antibodies recognition. MF and SS are both made of memory T lymphocytes. All leucocytes express CD45 and all T-lymphocytes express CD3. Sézary cells are usually CD3^+^, CD4^+^, or CD8^−^. Loss of expression of CD7 or CD26 is a criterion in favor of MF/SS [12].

Latest recommendations regarding treatment of MF and SS have been published in 2017 by the European Organization for Research and Treatment of Cancer—Cutaneous Lymphoma Task Force (EORTC-CLTF [9]). Stage is determined by the affected skin surface (T), the presence of any clinical lymphadenopathy and whereas histology confirms that it is lymphoma involvement (N) [13], visceral tumors (M) and blood invasion (B). This “B” criteria rely on the flow cytometry analysis since B0, B1 and B2 depends on the number of CD4^+^CD7^−^ and CD4^+^CD26^−^ T-cell lymphocytes.

Advanced CTCL may be difficult to treat. First line options are photopheresis with/or systemic retinoids, alpha interferon, methotrexate, or bexarotene. Second line options are chemotherapy (gemcitabine, doxorubicin, romidepsin), radiotherapy, or immunotherapy such as mogamulizumab or alemtuzumab.

Mogamulizumab is a kappa-humanized IgG1-type monoclonal antibody that specifically bonds to the CCR4 receptor involved in lymphocytes’ migration to organs, including the skin, leading to the depletion of these targeted cells. The CCR4 receptor is expressed on the surface of Sézary cells and is correlated with tumor proliferation [14].

The most frequent side effects of mogamulizumab are infusion-related reactions, drug rash, and diarrhea. It can induce auto-immune diseases or immunodepression by depletion of regulatory T-lymphocytes [15]. A MAVORIC study has shown that mogamulizumab improves progression-free survival (PFS) compared to vorinostat (a histone deacetylase inhibitor) [16].

Mogamulizumab (Poteligeo^®^, Kyowa Kirin Pharma, Tokyo, Japan) can be prescribed in MF/SS stages III to IV in the second line after a systemic treatment or in stages IB-II after two unsuccessful systemic treatments. It has been authorized in the US and France since 2018. Since MF and SS are rare and mogamulizumab is a new drug only used in a small portion of patients, we lack data on long-term efficiency and potential side effects in real-life data.

Our study aims to determine efficacy considering the median PFS of advanced CTCL with mogamulizumab. Secondary objectives were to consider tolerance and estimate delay until side effect appearance and if a profile of a responsive patient could be distinguished. 

## 2. Materials and Methods

Data on patients with advanced cutaneous T-cell lymphomas (*n* = 24) were collected until 1 September 2021, in 6 French university hospitals (Caen, *n* = 6; Rennes, *n* = 4; Nantes, *n* = 6; Strasbourg, *n* = 3; Tours, *n* = 3 and Amiens, *n* = 2). We excluded 3 patients in Nantes because they had already been included in the MAVORIC study and data was not available. Patients were studied since the beginning of treatment (first inclusion in November 2018) until they stopped mogamulizumab because of relapse or toxicity. For those still treated by mogamulizumab end point was 1 September 2021.

The study complies with the ethical standards resulting from the Declaration of Helsinki. This observational study did not involve the patients differently than their usual management (reuse of their health data). Oral information was delivered to the patients and none of them were opposed to it.

All data was collected from medical files filed by the referring dermatologist. Adverse events (AEs) were defined by unfavorable symptoms or diseases temporally associated with the use of mogamulizumab that may be considered related to this treatment by local pharmacovigilance surveys (even when there was no proof nor investigation). The grading of severe adverse events (grade III–IV) was determined by treating physicians (dermatologists) using the Common Terminology Criteria for Adverse Events (CTCAE) [17]. All baseline characteristics of patients are summed up in Table 1.

Responder patients were patients evolving in full response, partial response, or stability. Non-responder patients were patients evolving to relapse or death. Full response was defined by absence of Sézary cells in blood and absence of skin lesions of mycosis fungoides. Partial response was defined by a significant decrease in Sézary cells or skin lesions. Stability was when present Sézary cells of skin lesions stood in place with no new lesion nor increase in Sézary count. Progression was defined by extension of skin lesions, increase in Sézary count, appearance of lymphadenopathy, or visceral involvement. Death group defined patients who died while or shortly after being treated by mogamulizumab.

### Statistical Analysis

Quantitative characteristics were described using means with their standard deviation in case of normal distribution and using medians with their first and third quartiles otherwise. Normality was tested using Shapiro–Wilk tests. Means were compared according to mogamulizumab response status using Student’s *t*-tests with testing of the equality of variances hypothesis. Medians were compared according to mogamulizumab response status using Wilcoxon–Mann–Whitney tests. Qualitative characteristics were described using numbers and percentages and compared according to mogamulizumab response status using either Khi-2 or Fisher exact tests.

Patient characteristics were compared before and after the introduction of mogamulizumab using Wilcoxon signed rank tests for quantitative variables and either McNemar or Fisher exact tests for qualitative variables. Survival curves were estimated by the Kaplan–Meier method. All analyses were performed with SAS 9.4 software (SAS Manufaturinc, Inc., Corona, CA, USA). The statistical significance level was set at 5%.

## 3. Results

We reported 16/21 SS and 5/21 MF. The median time of follow-up was 11.6 months.

Patients’ evolutions are presented in Table 2. We reported one patient with visceral progression and seven had new lymphadenopathy, but adenectomy and histology were only performed in two patients. The occurrence of progression or death is modelized in Figure 1. Progression-free survival was estimated at 22 months. Time of response to mogamulizumab before progression or death and time of follow up are presented in Table 2.

When comparing patients’ database with opposing responders and non-responders (Table 1), we did not report any significant differences (*p* < 0.05). There was a tendency (*p* < 0.20) regarding sex, delay between diagnosis and introduction of mogamulizumab, ECOG status, therapeutic line, and affected skin surface at introduction. 

We did not compare immunophenotypic profiles because of the following lack of data and potential risk of alpha inflation: biological complete immunophenotyping (including presence or loss of expression of markers CD2, CD3, CD4, CD5, CD7, CD8, and CD26) was performed at some point between diagnosis and mogamulizumab’s introduction in 13/21 patients. 

Cutaneous immunophenotyping was performed in 20/21 patients.

Biological immunophenotyping was performed at some point after mogamulizumab’s introduction in 11/21 patients. One center performed on all its patients’ systematic biological immunophenotyping every trimester, at diagnosis and before each therapeutic modification. The frequency of cutaneous and biological immunophenotyping in other centers was heterogeneous.

Of the 21 patients, we observed five deaths (due to severe infections, strokes, and multivisceral dysfunction).

In total, 20 patients presented adverse events, of which 10 were severe, i.e., grade III–IV AEs. The occurrence of adverse events is given in Figure 2. The median time between the introduction of mogamulizumab and the first adverse event was 21 days. The delay between the introduction of mogamulizumab and the occurrence of the first severe adverse event was longer in responder patients than in non-responder patients (*p* = 0.089) (Table 3; Figure 3).

Clinical and biological features before and after mogamulizumab treatment are given in Table 4. The number of Sézary cells and the CD4/CD8 ratio significantly decreased (*p* < 0.05). LDH level, affected skin surface, and lymphadenopathy were not significantly modified.

## 4. Discussion

Our study shows that the PFS of patients treated with mogamulizumab is estimated at 22 months. As far as we know, this is the first multicentric real-life data settings study. AEs induced by mogamulizumab appear precociously, most of them within the first month of treatment.

In our study, the PFS of 22 months reflects the following clinical efficacy, mostly superior to that in other studies: 15.1 months in patients with acute CTCL in Japan [18], 7.7 months in the worldwide MAVORIC study [16]. Biological efficacy is demonstrated by a significative drop of Sézary cells count and CD4/CD8 ratio which are markers of blood involvement and predictive prognosis markers [9,19]. Nevertheless, significant improvement in skin surface involvement was not observed, and quality of life was not assessed.

Our population is composed of advanced CTCL with multiple anterior therapeutic lines, in accordance with marketing authorization. Contrary to the MAVORIC study [16], we had no other exclusion criterion, particularly concerning other comorbidities. Therefore, we strongly limit selection bias. The MAVORIC study [16], with a large recruitment of 184 patients treated by mogamulizumab, recruited most patients in reference centers with physicians being experts in CTCL. It excluded patients with an ECOG score of >1 and those with unresolved AEs from previous therapeutic lines. Yet, in everyday practice, mogamulizumab is used in advanced CTCL, influencing ECOG status and after multiple treatment failures, sometimes directly because of AEs contraindicating previous treatments.

This study is indeed innovative in pointing out the interest of mogamulizumab in real life to treat advanced CTCL with therapeutic failure. It is particularly interesting since some countries or hospitals do not finance this treatment anymore, which is justified by the high cost of these new immunotherapies. This data tends to confirm mogamulizumab’s place in the therapeutic approach of CTCL.

Most AE’s appearing within the first perfusions suggests a dose-dependent mechanism rather than cumulative.

We report 10/21 grade III–IV AEs (47%), which is comparable to the MAVORIC study (41% severe AEs in mogamulizumab but also in Vorinostat).

AEs appearing later in responders is significant but are to be considered with precaution. Classification bias is possible since some AEs may have caused death and then categorized patients as non-responders, making it impossible to know if mogamulizumab would have been efficient if used long enough. Moreover, we consider all AEs during the period of treatment even if there is no proof nor pharmacovigilance notice in favor of mogamulizumab’s causality. Some of them may be due to previous therapeutic lines.

In this way, our study is more binding than MAVORIC since every AE occurring during mogamulizumab has been considered accountable to the molecule, even if pharmacovigilance has expressed some reserves about accountability or when pharmacovigilance notice was lacking. We collect all AEs with mogamulizumab, and therefore, we overestimate the accountability of mogamulizumab de facto. This suggests that safety data should be even better than what we report in our study.

With flow cytometry’s development and immunophenotyping precision, it seemed interesting to see if a clone profile stood out from responders and non-responders so that we could narrow mogamulizumab’s indication. Systematic and precise immunophenotyping follow-up was performed only in one center. Therefore, the scarcity of data does not allow statistical evaluation of our 21 patients. Nowadays, flow cytometry tends to be the reference in terms of diagnosis and prognosis. Guidelines have recently been published to harmonize practices and facilitate the conduct of informative and comparable clinical trials [20]. This will probably incite physicians to perform extensive immunophenotyping at least at diagnosis and at the start of each therapeutic line. This will facilitate and empower future studies.

Our recruitment of 21 patients and the lack of randomization were an obstacle to getting significant results. Larger studies will be necessary to confirm our findings and strengthen statistical power. Nevertheless, this first multicentric real-life data study supports the fact that mogamulizumab is efficient in 13/21 patients presenting late stage CTCL. Previous studies demonstrated that it is a safe drug [21,22,23], which makes it interesting in CTCL that are not responding to classical treatments.

CTCL is difficult to treat; new treatments are in development. Mogamulizumab was one of the first new CTCL treatments and many others are being studied, such as monoclonal antibodies KIR3DL2 (lacutamab) [24], anti-PD1 (pembrolizumab, nivolumab) [25,26], anti-PDL1 (atézolizumab), anti-CD47 [27], and anti-CD70 [28,29]. This brings hope to improve advanced CTCL prognosis in the future.

## 5. Conclusions

Our study suggests that mogamulizumab can give patients with advanced refractory CTCL a substantial PFS, estimated at 22 months. The long-term safety of mogamulizumab was determined to be acceptable since we reported 47% grade III–IV AEs in our cohort of vulnerable patients. 

No other study using real-life data has been performed to investigate the AEs of mogamulizumab. Further multicentric studies with a wider recruitment of patients should be performed to confirm our findings and obtain more powerful results.

## Figures and Tables

**Figure 1 cancers-14-01659-f001:**
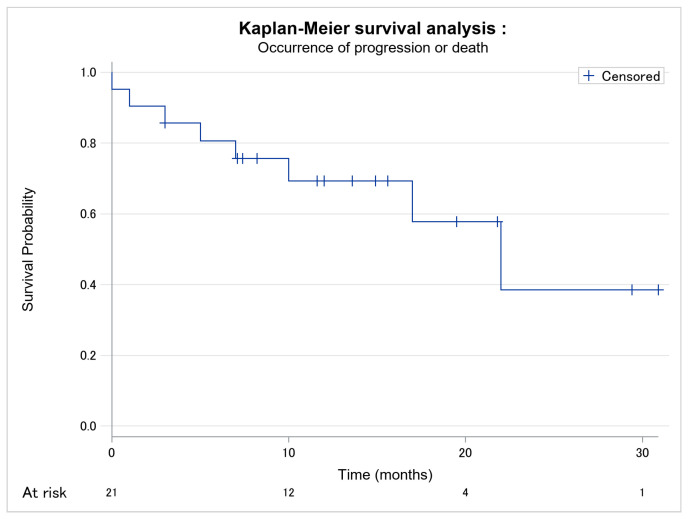
Kaplan–Meier survival analysis of occurrence of progression or death among the 21 patients. No missing data.

**Figure 2 cancers-14-01659-f002:**
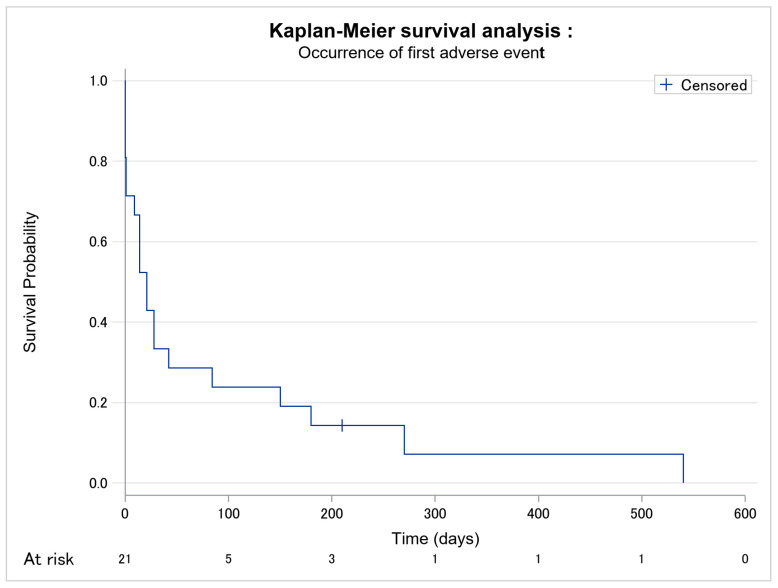
Kaplan–Meier survival analysis of occurrence of first all-grade adverse event among the 21 patients. No missing data.

**Figure 3 cancers-14-01659-f003:**
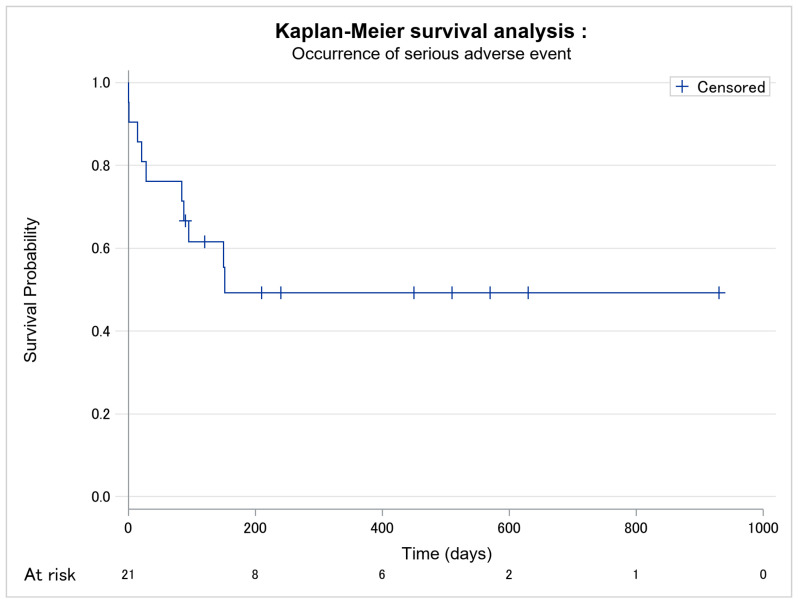
Kaplan–Meier survival analysis of occurrence of severe grade III–IV adverse event among the 21 patients. No missing data.

**Table 1 cancers-14-01659-t001:** Patient characteristics and comparison between responders and non-responders to mogamulizumab. The *p*-value obtained using Khi-square or Fisher exact test for qualitative variables and using Student’s t-test or Mann–Whitney U test for quantitative variables, according to their distribution.

Basic Features	Population (*n* = 21)	Responders (*n* = 13)	Non-Responders (*n* = 8)	*p*-Value
Sex (*n*, %)				0.067
Male	12 (57.1)	5 (38.5)	7 (87.5)	-
Female	9 (42.9)	8 (61.5)	1 (12.5)	-
Age (moy ± δ)	68.0 ± 10.7	70.0 ± 11.9	64.8 ± 7.9	0.286
Smoker (*n*, %) (*n* = 18 [12/6])	4 (22.2)	2 (16.7)	2 (33.3)	0.569
Alcohol consumption (*n*, %) (*n* = 18 [12/6])	4 (22.2)	2 (16.7)	2 (33.3)	0.569
Lymphoma type (*n*, %)				
Sézary	16 (76.2)	10 (76.9)	6 (75.0)	1.000
Mycosis fungoïde	5 (23.8)	3 (23.1)	2 (25.0)	
Delay from diagnosis to mogamulizumab introduction (in months) [median, Q1–Q3]	32.0 [22.0–87.0]	25.0 [15.0–87.0]	44.5 [33.5–107.0]	0.169
T4-stage at mogamulizumab introduction (*n*, %)	12 (57.1)	7 (53.9)	5 (62.5)	1.000
ECOG status at mogamulizumab introduction (*n*, %)				0.194
0	10 (47.6)	5 (38.5)	5 (62.5)	0.387
1	6 (28.6)	4 (30.8)	2 (25.0)	1.000
2	4 (19.1)	4 (30.8)	0 (0.0)	0.131
3	1 (4.8)	0 (0.0)	1 (12.5)	-
LDH level at mogamulizumab introduction (UI/L) [median, Q1–Q3] (*n* = 17 [10/7])	299.0 [211.0–365.0]	260.0 [202.0–358.0]	310.0 [211.0–379.0]	0.591
LDH level superior to 245 UI/L at mogamulizumab introduction (*n*, %) (*n* = 18 [10/8])	10 (55.6)	5 (50.0)	5 (62.5)	0.664
Number of Sézary cells at mogamulizumab introduction (g/L) [median, Q1–Q3] (*n* = 20 [12/8])	0.88 [0.02–6.66]	2.85 [0.20–9.50]	0.39 [0.003–3.15]	0.296
CD4/CD8 ratio before mogamulizumab introduction [median, Q1–Q3] (*n* = 17 [11/6])	9.6 [3.3–88.0]	9.6 [3.0–89.0]	8.4 [3.3–40.0]	0.580
CD4/CD8 ratio >10 before mogamulizumab introduction (*n*, %) (*n* = 17 [11/6])	7 (41.2)	5 (45.5)	2 (33.3)	1.000
Number of therapeutic lines [median, Q1–Q3]	4.0 [3.0–5.0]	3.0 [3.0–5.0]	5.0 [4.5–6.0]	0.070
Patch type cutaneous before introduction of mogamulizumab (*n*, %)	8 (38.1)	6 (46.2)	2 (25.0)	0.400
Plaque type cutaneous lesions (infiltrated) before introduction of mogamulizumab (*n*, %)	7 (33.3)	5 (38.5)	2 (25.0)	0.656
Tumor type cutaneous lesions before introduction of mogamulizumab (*n*, %)	2 (9.5)	1 (7.7)	1 (12.5)	1.000
Presence of any adenopathy before introduction of mogamulizumab (*n*, %)	12 (57.1)	7 (53.9)	5 (62.5)	1.000
Cutaneous area affected before introduction of mogamulizumab (*n*, %)				0.110
0%	1 (4.8)	0 (0.0)	1 (12.5)	0.381
<10%	2 (9.5)	2 (15.4)	0 (0.0)	0.505
10–50%	1 (4.8)	0 (0.0)	1 (12.5)	0.381
5–80%	4 (19.1)	4 (30.8)	0 (0.0)	0.131
>80%	13 (61.9)	7 (53.9)	6 (75.0)	0.400

**Table 2 cancers-14-01659-t002:** Descriptive table of patient’s evolution with mogamulizumab. *p*-value obtained using Fisher exact test for qualitative variables and using Mann–Whitney U test for quantitative variables.

Basic Features	Population (*n* = 21)	Responders (*n* = 13)	Non-Responders (*n* = 8)	*p*-Value
**Evolution (*n*, %)**				
Full response	4 (19.1)	4 (30.8)	-	-
Partial response	8 (38.1)	8 (61.5)	-	-
Stability	1 (4.8)	1 (4.8)	-	-
Progression	3 (14.3)	-	3 (37.5)	-
Death	5 (23.8)	-	5 (62.5)	-
**Time of follow-up (in months) [median, Q1–Q3]**	11.6 [7.0–17.0]	13.6 [8.2–19.5]	6.0 [2.0–13.5]	-
**Time of response to Moga before progression or death (in months) [median, Q1–Q3] (*n* = 8)**	-	-	6.0 [2.0–13.5]	-
Before progression (*n* = 3)	-	-	17.0 [3.0–22.0]	-
Before death (*n* = 5)	-	-	5.0 [1.0–7.0]	
**Responder’s time of follow-up in months) [median, Q1–Q3] (*n* = 13)**	-	13.6 [8.2–19.5]	-	-
Full response (*n* = 4)	-	22.5 [15.3–30.2]	-	-
Partial response (*n* = 8)	-	9.9 [7.3–16.6]	-	-
Stability (*n* = 1)	-	12.0 [12.0–12.0]	-	-
Visceral progression (*n*, *%*)	1 (4.8)	-	1 (12.5)	-
Presence of one or several adenopathy (*n*, *%*)	5 (23.8)	-	5 (62.5)	-
**LDH-level at endpoint (UI/L) [median, Q1–Q3] (*n* = 14 [9/5])**	268.0 [201.0–363.0]	287.0 [221.0–323.0]	249.0 [201.0–549.0]	0.790
LDH-level superior 245 UI/L at end-point (*n*, *%*) (*n* = 14 [9/5])	8 (57.1)	5 (55.6)	3 (60.0)	1.000
**Number of Sézary cells (progression or endpoint, g/L) [median, Q1–Q3] (*n* = 20 [13/7])**	0.00 [0.00–0.80]	0.00 [0.00–0.65]	0.55 [0.00–3.13]	0.165
**CD4/CD8 ratio at endpoint [median, Q1–Q3] *n* = 12 [8/4])**	3.8 [1.1–7.2]	4.2 [1.1–7.2]	3.3 [1.1–31.6]	1.000
CD4/CD8 ratio >10 at end-point (*n*, *%*) (*n* = 13 [9/4])	2 (15.4)	1 (11.1)	1 (25.0)	1.000
**Total number of lymphocytes (progression or endpoint, g/L) [median, Q1–Q3]**	0.99 [0.70–2.14]	0.99 [0.80–2.08]	1.20 [0.65–3.67]	0.800
**Clinical presence of skin lesions (progression or end-point (*n, %*)**	18 (85.7)	10 (76.9)	8 (100.0)	0.257

**Table 3 cancers-14-01659-t003:** Descriptive table of adverse events during mogamulizumab treatment. *p*-value obtained using Fisher exact test for qualitative variables and using Mann–Whitney U test for quantitative variables.

Basic Features	Population (*n* = 21)	Responders (*n* = 13)	Non-Responders (*n* = 8)	*p*-Value
Adverse event occurrence (*n*, *%*)	20 (95.2)	13 (100.0)	7 (87.5)	0.381
Delay between mogamulizumab introduction and first adverse event (days) [median, Q1–Q3]	21.0 [1.0–84.0]	21.0 [14.0–84.0]	18.5 [0.5–126.0]	0.913
Occurrence of severe adverse event (*n*, *%*)	10 (47.6)	5 (38.5)	5 (62.5)	0.387
Delay between mogamulizumab introduction and first severe adverse event (days) [median, Q1–Q3]	120.0 [84.0–450.0]	152.0 [120.0–450.0]	88.5 [14.5–152.5]	0.089
Number of adverse events per patient since mogamulizumab [median, Q1–Q3]	2.0 [1.0–3.0]	2.0 [1.0–3.0]	1.5 [1.0–2.5]	0.515

**Table 4 cancers-14-01659-t004:** Comparative table: before mogamulizumab-under mogamulizumab. *p*-value obtained using Fisher exact test for qualitative variables and Wilcoxon signed ranks test for quantitative variables.

	Population (*n* = 21)			Responders (*n* = 13)			Non-Responders (*n* = 8)		
	before Moga	with Moga	*p*-Value	before Moga	with Moga	*p*-Value	before Moga	with Moga	*p*-Value
LDH-level (UI/L) [median, Q1–Q3]	299.0 [211.0–365.0]	268.0 [201.0–363.0]	0.671	260.0 [202.0–358.0]	287.0 [221.0–323.0]	1.000	310.0 [211.0–379.0]	249.0 [201.0–549.0]	0.813
LDH-level superior to 245 UI/L (*n*, *%*)	10 (55.6)	8 (57.1)	1.000	5 (50.0)	5 (55.6)	1.000	5 (62.5)	3 (60.0)	1.000
Number of Sézary cells(g/L) [median, Q1–Q3]	0.88 [0.02–6.66]	0.00 [0.00–0.80]	0.004	2.85 [0.20–9.50]	0.00 [0.00–0.65]	0.006	0.39 [0.003–3.15]	0.55 [0.00–3.13]	0.438
CD4/CD8 ratio [median, Q1–Q3]	9.6 [3.3–88.0]	3.8 [1.1–7.2]	0.001	9.6 [3.0–89.0]	4.2 [1.1–7.2]	0.016	8.4 [3.3–40.0]	3.3 [1.1–31.6]	0.125
CD4/CD8 >10 (*n*, *%*)	7 (41.2)	2 (15.4)	0.455	5 (45.5)	1 (11.1)	1.000	2 (33.3)	1 (25.0)	1.000
Presence of adenopathy (*n*, *%*)	12 (57.1)	8 (38.1)	0.367	7 (53.9)	3 (23.1)	1.000	5 (62.5)	5 (62.5)	0.464
Affected cutaneous area (*n*, *%*)	20 (95.2)	18 (85.7)	1.000	13 (100.0)	10 (76.9)	-	7 (87.5)	8 (100.0)	-

## Data Availability

The data presented in this study are available on request from the corresponding author.

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
