# Peer review of "Impact of Mogamulizumab in Real-Life Advanced Cutaneous T-Cell Lymphomas: A Multicentric Retrospective Cohort Study"

_cancers, 2022, doi:10.3390/cancers14071659_

Round 1

Reviewer 1 Report

Reviewer comments:

Comments to the Author

The present study demonstrates the use of real-life data to investigate adverse events of Mogamulizumab.

This manuscript is for the most part well written and discussed the recent studies. However, to increase the readability and comprehensiveness, authors are advised to incorporate some suggestions.

Minor criticisms

  • Authors should provide the full meaning of the abbreviations when they come first throughout the manuscript.

  • Please provide the detail figure legends for the figure such as Figure 1, Figure 2 and Figure 3. Authors should include the statistical analysis with the number of patients or samples included.

  • Whether protocol to include advanced cutaneous T-cell lymphomas patients were approved? Include the protocol approval number with the statement in the Methods section.

  • It is confusing if authors have performed any immunophenotyping. Please clarify this. No data was provided relate to immunophenotyping that was performed in one Center. Please provide the data. Also, please include the flow cytometry and immunophenotyping precision, which were applied in this manuscript as discussed in the discussion.

Author Response

Dear reviewer,

First, we would like to express our gratitude for the time that you dedicated to revise our manuscript. We are glad that you understood our work and that you find some interest in this subject.

We considered all your suggestions carefully and made the appropriate modifications. We hope that you will find this revised version valuable of publication.

Kind regards,

Marie Jouandet, dermatologist resident.

Minor criticisms

  • Authors should provide the full meaning of the abbreviations when they come first throughout the manuscript.

We thank you for pointing this out. We included the full meaning of the abbreviations into the abstract and the small summary.

  • Please provide the detail figure legends for the figure such as Figure 1, Figure 2 and Figure 3. Authors should include the statistical analysis with the number of patients or samples included.

We included all patients in the three Kaplan-Meyer analysis. We completed the figure legends in this way.

We replaced comas by dots in tables and we deleted the column “test” since it was described in the method section. We completed the table’s title.

  • Whether protocol to include advanced cutaneous T-cell lymphomas patients were approved? Include the protocol approval number with the statement in the Methods section.

We appreciate your insightful suggestion and agree that it would have been pertinent for an international publication to have an inclusion protocol. However, this study does not have a registration number in accordance with French legislation.

The study complies with the ethical standards resulting from the declaration of Helsinki. It is an observational study which did not involve the patients differently from their usual management (reuse of their health data). Oral information was delivered to the patients and none of them were opposed to it.

We included this paragraph in the Methods section.

  • It is confusing if authors have performed any immunophenotyping. Please clarify this. No data was provided relate to immunophenotyping that was performed in one Center. Please provide the data. Also, please include the flow cytometry and immunophenotyping precision, which were applied in this manuscript as discussed in the discussion.

We added a paragraph (Page 9 line 231) to clarify data about immunophenotyping implementation.

Reviewer 2 Report

Dear Authors, I read the article with great interest . Treatment of advanced Mycosis fungicides and Sezary syndrome has been challenging. You present the real data of mogamulizumab  treatment. Materials and methods have been adequately described, results  -  clearly presented and conclusions have been supported by them.   I found the article valuable and worth of publication for sure. Just  check the English well - especially in tables ( there is Lymphome, French name of MF etc).

Yours sincerely

Author Response

Dear reviewer,

We would like to express our gratitude for taking the time to assess our manuscript.

It was a great pleasure to read your commentary about our work. We are glad that you understood the purpose of this paper, and we wish to thank you for encouraging its publication.

We had the English language corrected by a multilingual physician. We changed the comas to dots within the numbers of tables.

We hope that the revised version will suit you.

Kind regards,

Marie Jouandet, dermatologist resident.